# A *PRPH* splice-donor variant associates with reduced sural nerve amplitude and risk of peripheral neuropathy

Gyda Bjornsdottir[1], Erna V. Ivarsdottir [1,2], Kristbjorg Bjarnadottir[1], Stefania Benonisdottir[1], Sandra Sif Gylfadottir[3], Gudny A. Arnadottir [1], Rafn Benediktsson [4,5], Gisli Hreinn Halldorsson [1], Anna Helgadottir [1], Adalbjorg Jonasdottir[1], Aslaug Jonasdottir[1], Ingileif Jonsdottir[1,4], Anna Margret Kristinsdottir[1], Olafur Th. Magnusson[1], Gisli Masson[1], Pall Melsted[1,2], Thorunn Rafnar [1], Asgeir Sigurdsson[1], Gunnar Sigurdsson[1,4,5], Astros Skuladottir[1], Valgerdur Steinthorsdottir [1], Unnur Styrkarsdottir [1], Gudmundur Thorgeirsson[1,4,5], Gudmar Thorleifsson[1], Arnor Vikingsson[5], Daniel F. Gudbjartsson [1,2], Hilma Holm[1,4], Hreinn Stefansson [1], Unnur Thorsteinsdottir[1,4], Gudmundur L. Norddahl[1], Patrick Sulem [1], Thorgeir E. Thorgeirsson[1] & Kari Stefansson[1,4]

Nerve conduction (NC) studies generate measures of peripheral nerve function that can reveal underlying pathology due to axonal loss, demyelination or both. We perform a genome-wide association study of sural NC amplitude and velocity in 7045 Icelanders and find a low-frequency splice-donor variant in *PRPH* (c.996+1G>A; MAF = 1.32%) associating with decreased NC amplitude but not velocity. *PRPH* encodes peripherin, an intermediate filament (IF) protein involved in cytoskeletal development and maintenance of neurons. Through RNA and protein studies, we show that the variant leads to loss-of-function (LoF), as when over-expressed in a cell line devoid of other IFs, it does not allow formation of the normal filamentous structure of peripherin, yielding instead punctate protein inclusions. Recall of carriers for neurological assessment confirms that from an early age, homozygotes have significantly lower sural NC amplitude than non-carriers and are at risk of a mild, early-onset, sensory-negative, axonal polyneuropathy.

[1] deCODE Genetics/Amgen, Inc., 101 Reykjavik, Iceland. [2] School of Engineering and Natural Sciences, University of Iceland, 101 Reykjavik, Iceland. [3] Danish Pain Research Center/Aarhus University Hospital, 8000 Aarhus, Denmark. [4] Faculty of Medicine, University of Iceland, 101 Reykjavik, Iceland. [5] Landspitali—The National University Hospital of Iceland, 101 Reykjavik, Iceland. Correspondence and requests for materials should be addressed to G.B. (email: gyda.bjornsdottir@decode.is) or to K.S. (email: kari.stefansson@decode.is)

Peripheral neuropathy (PN) is a term applied to a diverse group of debilitating and often painful diseases of peripheral nerves, for which limited treatment is available[1,2]. Over 100 types of PN have been described, with various clinical signs and symptoms depending on nerves affected (motor, sensory, autonomic), distribution of disease (mono-, focal-, poly-neuropathy), temporal progression (acute, chronic), etiology, and whether the primary pathology involves axonal loss or myelin degeneration[1,2]. PN is a considerable public health problem with prevalence estimated at ~2%, rising to ~8% after the age of 55 years[2,3]. Etiological profiles differ between countries, with inflammatory neuropathy secondary to infectious diseases most prevalent in developing countries, whereas in affluent societies, PN is more commonly a consequence of diabetes or other metabolic disorders, alcohol abuse, or cytotoxic drugs[2,3]. Additionally, several rare, hereditary forms of PN exist, including Charcot-Marie-Tooth (CMT) and other hereditary sensory and autosomal neuropathies (HSAN)[4–6]. Over 70 genes are linked to CMT and HSAN, their functions providing important insights into complex pathogeneses involving both axons and their myelin sheaths[4–6]. To date, three sequence variants are reported in the genome-wide association study (GWAS) catalog (URLs) to associate ($P < 1 \times 10^{-8}$) with non-hereditary forms of PN; two with PN secondary to anti-retroviral therapy in HIV patients (in *IL2RA* and *CXCL12*)[7], and one with PN induced by vincristine therapy for leukemia (in *CEP72*) in children[8] and in adults[9]. However, these findings are based on small samples.

Nerve conduction (NC) studies probe electrophysiological properties of nerves and are used to aid in the diagnosis and classification of PN[1,2,4,10]. For example, CMT is divided into demyelinating (CMT1) and axonal (CMT2) forms on the basis of NC studies[4]. In general, a marked reduction in NC velocity indicates demyelination, whereas reduced amplitude with relative preservation of velocity indicates axonal degeneration[10–12]. While the heritability of peripheral NC velocity is high (77%)[13], GWAS of NC measures has not been reported.

Here, we describe our search for sequence variants associating with sural NC measures in the Icelandic population that results in discovery of a LoF splice-donor variant in *PRPH*, associating with reduced amplitude of the sural nerve action potential (SNAP), but not with sural NC velocity (SNCV). By functional and clinical follow-up studies, we describe the biological effect of this variant, providing the mechanistic basis of a novel, early-onset, predominantly sensory and axonal PN.

## Results

**Sural nerve conduction traits correlate with body measures.** The sample for this study is based on NC measures of the sural nerve in 7045 Icelanders; 6879 with SNAP; and 6979 with SNCV. Sural NC was measured in the right lower calf region using DPNCheck® (Neurometrix®, Methods). NC measures were obtained as part of comprehensive phenotyping of a general population sample enriched for homozygous carriers of rare and potentially high impact mutations[14]; the deCODE Health Study ($N = 6488$), and subjects recruited as controls for a study on genetics of chronic pain ($N = 557$). The two sample sets did not differ in terms of prevalence of chronic pain (defined as pain on most days for ≥3 months); hence, NC measures from both recruitments were combined. The total sample's demographics and descriptive data are shown in Table 1.

As age and various physical features are known to affect NC results[15,16], we studied Pearson correlations between SNAP on the log scale (due to skewness) and SNCV, demographics, height, body mass index (BMI), and leg fat mass measured by dual-energy X-ray absorptiometry (DEXA, Methods), since adipose tissue over the measured sural nerve has insulating effects. We find that SNAP and SNCV are correlated ($r = 0.27$, $P = 5.1 \times 10^{-113}$), that both SNAP and SNCV decrease with age ($-0.02 \log(\mu V)$ per year, $P < 1.0 \times 10^{-308}$ and $-0.11$ m s$^{-1}$ year$^{-2}$, $P = 1.4 \times 10^{-100}$, respectively), and males have higher SNAP and lower SNCV than females (2.24 vs 2.17 log ($\mu V$), $P = 2.7 \times 10^{-5}$; 50.0 vs 54.0 m s$^{-1}$, $P = 3.1 \times 10^{-166}$, respectively). Performing multiple regression analyses for both SNAP and SNCV with sex, age, height, BMI, and leg fat mass as covariates reveals that the difference in SNCV between sexes is driven solely by the effect of these anthropometric measures (Effect = 0.07 m s$^{-1}$, $P = 0.80$), while SNAP is still significantly different between the sexes after adjusting for covariates (Effect = $-0.10 \log(\mu V)$, $P = 1.4 \times 10^{-4}$), a finding that has not been reported before[15,16]. Increased height associates with both decreased SNAP (Effect = $-0.01 \log(\mu V)$, $P = 4.0 \times 10^{-32}$) and SNCV (Effect = $-0.25$ m sec$^{-1}$, $P = 1.2 \times 10^{-87}$), while increased leg fat mass associates with decreased SNAP (Effect = $-0.08 \log(\mu V)$, $P = 3.5 \times 10^{-37}$) and increased SNCV (Effect = 0.39 m s$^{-1}$, $P = 1.9 \times 10^{-8}$). These and additional correlations between NC measures and blood levels of several biomarkers known to associate with PN[17]; glucose, vitamin B$_{12}$, and thyroid-stimulating hormone (TSH) are shown in Supplementary Fig. 1 and Supplementary Tables 1 and 2.

**A *PRPH* variant associates with reduced SNAP but not SNCV.** To search for sequence variants associating with SNAP and SNCV, we analyzed 37.6 million variants detected through whole-genome sequencing of 28,075 Icelanders, and imputed into 155,250 chip-typed individuals and their relatives (methods)[18]. Prior to association testing, we rank-based inverse standard normal transformed SNAP and SNCV measures separately for each sex and adjusted SNCV for age and height and SNAP for age, height, and leg fat mass.

Under the additive model, we identified a genome-wide significant association with SNAP at chromosome 12q13.12. The signal is represented and fully explained by rs73112142 within peripherin (*PRPH*) (NM_006262.3: c.996+1G>A) (Fig. 1). The minor allele, rs73112142-A (MAF = 1.32%), is associated with decreased SNAP ($\beta = -0.49$ SD, equivalent to $\beta = -0.26 \log(\mu V)$, $P = 1.14 \times 10^{-11}$). The likelihood ratio test was performed in all genome-wide associations. The variant

**Table 1 Demographics and descriptive statistics for the GWAS sample, $N = 7045$**

|  | Males | Females | P* | Total |
|---|---|---|---|---|
| N (%) | 3140 (44.6) | 3905 (55.4) | - | 7045 (100) |
| Age, M (SD) | 54.4 (15.1) | 54.9 (14.4) | 0.51 | 55.7 (14.8) |
| SNAP, M (SD) | 11.3 (7.2) | 10.3 (6.2) | $6.2 \times 10^{-11}$ | 10.7 (6.6) |
| SNCV, M (SD) | 50.0 (5.7) | 54.0 (6.0) | $1.8 \times 10^{-166}$ | 52.2 (6.2) |
| Chronic pain, N (%) | 181 (5.8) | 376 (9.6) | $1.7 \times 10^{-9}$ | 557 (7.9) |

*N = 6879 with SNAP (sural nerve action potential, amplitude, μV) and N = 6979 with SNCV (sural nerve conduction velocity, m s$^{-1}$), chronic pain (≥3 months, by self-report). M mean, SD standard deviation *Significance tests: Chi-square test for chronic pain and two-sided t-test for age, SNAP, and SNCV*

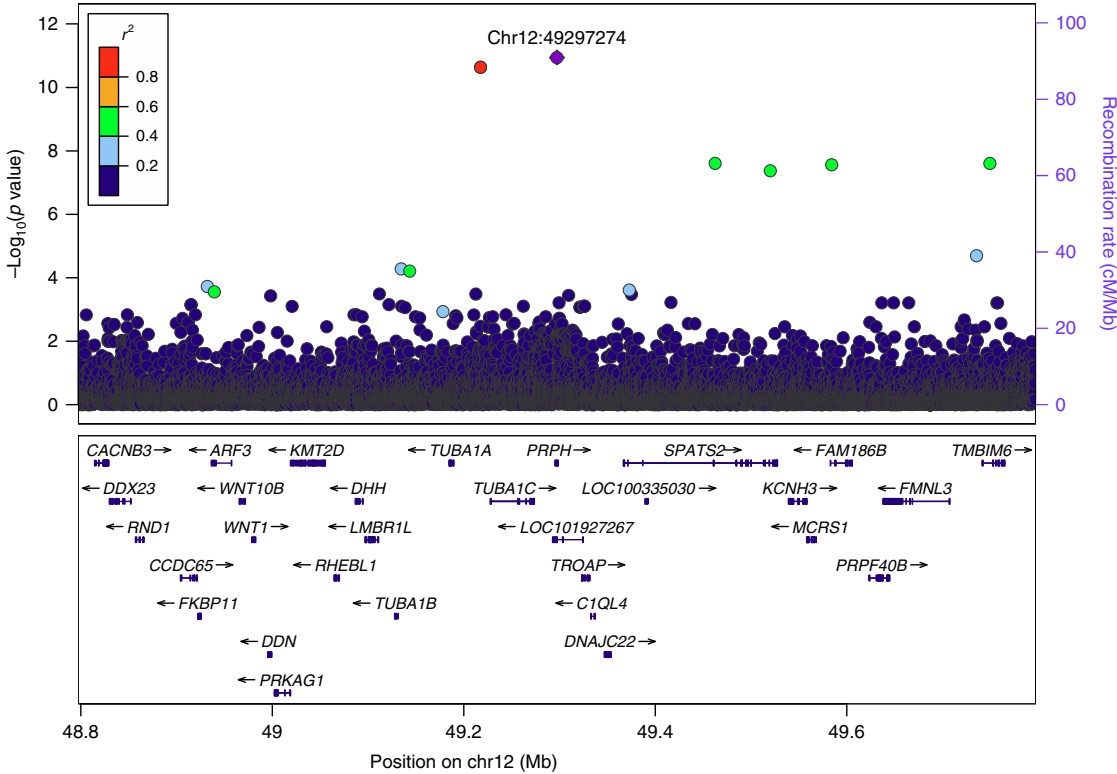

**Fig. 1** Regional association plot for the *PRPH* splice-donor variant association with SNAP. P values ($-\log_{10}$) of single-nucleotide polymorphism (SNP) associations with SNAP ($N = 6879$) are plotted against their positions at the 12q13.12 locus. SNPs are colored to reflect their linkage disequilibrium (LD) with rs73112142 (chr12:49297274, purple diamond) in the dataset. The only other marker in the vicinity is an intergenic, highly correlated variant rs532888409 ($r^2 = 0.93$, $D' = 0.97$). The right y-axis shows calculated recombination rates at the chromosomal location, plotted as solid gray lines (none in the shown region). Known genes in the region are shown underneath the plot, taken from the UCSC genes track in the UCSC Genome Browser (URLs). All positions are in NCBI Build 38 coordinates. A chi-squared test was used to calculate P values. Source data are provided in a Source Data file

**Table 2 Association of rs73112142-A with SNAP by genotype ($N = 6879$)**

| Tests | Effect<sup>S</sup> | CI<sup>S</sup> | $p^{\mathrm{S}}$ | $P_{\mathrm{het}}$ | Effect<sup>L</sup> | CI<sup>L</sup> | $p^{\mathrm{L}}$ | N (by gt) | $P_{\mathrm{het}}$ |
|---|---|---|---|---|---|---|---|---|---|
| Additive | −0.49 | (−0.63, −0.34) | $1.1 \times 10^{-11}$ | | −0.26 | (−0.33, −0.19) | $1.5 \times 10^{-12}$ | 10/172/6697 | |
| Males | −0.48 | (−0.67, −0.29) | $8.0 \times 10^{-7}$ | | −0.25 | (−0.37, −0.17) | $1.8 \times 10^{-7}$ | 7/88/2974 | |
| Females | −0.51 | (−0.72, −0.30) | $1.5 \times 10^{-6}$ | 0.83 | −0.27 | (−0.34, −0.16) | $3.0 \times 10^{-7}$ | 3/84/3723 | 0.78 |
| Recessive | −1.54 | (−2.16, −0.90) | $1.7 \times 10^{-6}$ | | −0.81 | (−1.12, −0.48) | $9.5 \times 10^{-7}$ | 10/6869 | |
| Males | −1.55 | (−2.29, −0.81) | $4.2 \times 10^{-5}$ | | −0.82 | (−1.20, −0.44) | $2.3 \times 10^{-5}$ | 7/3062 | |
| Females | −1.51 | (−2.65, −0.37) | $9.7 \times 10^{-3}$ | 0.95 | −0.77 | (−1.38, −0.16) | 0.014 | 3/3807 | 0.89 |
| Ht vs NC | −0.42 | (−0.57, −0.26) | $5.9 \times 10^{-8}$ | | −0.23 | (−0.32, −0.15) | $1.8 \times 10^{-8}$ | 172/6697 | |
| Hm vs NC | −1.54 | (−2.16, −0.92) | $9.9 \times 10^{-7}$ | | −0.81 | (−1.13, −0.49) | $7.5 \times 10^{-7}$ | 10/6697 | |
| Hm vs Ht | −1.12 | (−1.76, −0.48) | $5.8 \times 10^{-4}$ | | −0.61 | (−0.98, −0.26) | $6.0 \times 10^{-4}$ | 10/172 | |

Shown are associations with standardized (<sup>S</sup>) and and log transferred (<sup>L</sup>) SNAP. Effect difference between males and females was tested for statistical heterogeneity with a Q-test ($P_{\mathrm{het}}$)[19] (Methods). Effect<sup>S</sup> in standard deviations (SD), Effect<sup>L</sup> in log($\mu$V), 95% confidence interval (CI), Additive model (Additive), Recessive model (Recessive), Homozygotes (Hm), Heterozygotes (Ht), Non-carriers (NC). Sample size (N) by genotype (gt)

has comparable effects on SNAP in males ($\beta = -0.48$ SD, $P = 8.0 \times 10^{-7}$) and females ($\beta = -0.51$ SD, $P = 1.5 \times 10^{-6}$), P value for the Q-test for heterogeneity[19], $P_{\mathrm{het}} = 0.83$ (Table 2). Overlapping with *PRPH*, on the opposite strand, is the uncharacterized non-coding RNA gene, *LOC101927267* (Fig. 1). However, in *PRPH*, the variant is a change from G to A at the first base of the splice-donor site in intron 5, thus disrupting the normal GT–AG splicing[20,21]. By contrast, the variant does not associate with SNCV ($\beta = -0.11$ SD, $P = 0.14$). We found no other genome-wide significant signals associated with SNAP or SNCV. (Manhattan plots in Supplementary Fig. 2).

The low-frequency *PRPH* variant, rs73112142-A, is present in other European ancestry populations with average European

MAF = 0.73%, but is less frequent in non-European populations (EXAC, URLs). We did not find adequately powered and comparably phenotyped samples from other populations, in which to replicate our findings. Through imputation methods[18], among 155,250 Icelanders we identified 39 homozygotes for rs73112142-A, of whom 26 were alive at the time of study. Of these, we had in our discovery sample, NC measurements for ten adults (20 years or older), unrelated at a genealogical distance of five meiotic events. Assessing the effect on SNAP per allele, we observed deviation from the additive model in that homozygotes have around three-fold lower SNAP than non-carriers (Table 2). In addition to the clear dose-dependent genotype effect on SNAP, we also observe an age-related decline in SNAP (Fig. 2 and

Supplementary Fig. 3). While the slope of age-related decline in SNAP is comparable for heterozygotes and non-carriers, we observed no age-related decline in homozygotes from 20 years on. Indeed, at 20 years of age, their SNAP levels are already low and comparable to that of 80-year-old non-carriers (Fig. 2).

**Defective splicing generates two truncated protein isoforms.**
*PRPH* encodes peripherin, a class-III intermediate filament (IF) protein, the only IF in its class that is expressed in neurons[22–24].

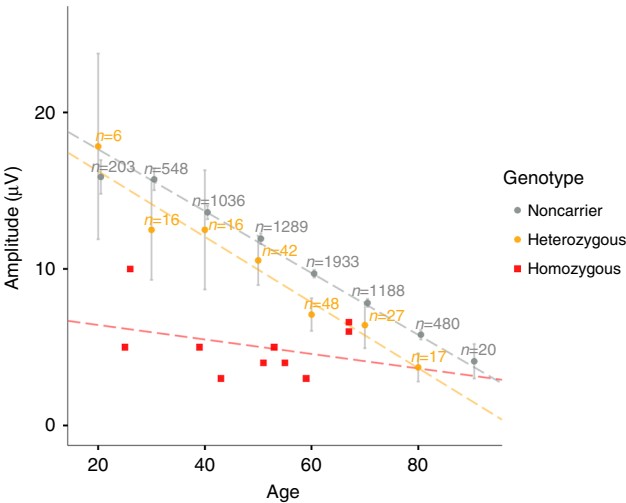

**Fig. 2** Sural nerve action potential (SNAP, µV) decreases with age ($N = 6879$). Shown is the mean SNAP per age group among non-carriers (gray dots) and heterozygotes (orange dots). Gray vertical lines represent 95% confidence intervals for the means. The red squares show individual values for the 10 homozygotes. The dotted lines represent results from a linear regression; SNAP (amplitude, µV) = $\beta 0 + \beta 1 \times$ Age, for each genotype group. Non-carriers: $N = 6697$; SNAP = $21.6 - 0.20 \times$ Age; $R^2 = 0.19$; $P = 1.2 \times 10^{-303}$. Heterozygotes: $N = 172$; SNAP = $20.4 - 0.21 \times$ Age; $R^2 = 0.29$; $P = 1.9 \times 10^{-14}$. Homozygotes: $N = 10$; SNAP = $7.3 - 0.05 \times$ Age; $R^2 = 0.12$; $P = 0.33$. Source data are provided in a Source Data file

Although the function of peripherin is not fully understood, it is involved in the development and maintenance of the axonal cytoskeleton and in axonal transport, along with other neurofilaments and microtubules[22–24]. Like other neuronal IFs, peripherin has a tripartite structure with non-helical amino- and carboxy-terminal regions (head and tail domains) flanking a central, highly conserved α-helical rod domain that is involved in the formation of coiled-coil filament structures (Fig. 3)[22–24]. In GTEX and Human Protein Atlas (URLs), *PRPH* shows significant RNA expression in both central- and peripheral nervous systems but is also expressed in other tissues, including adipose tissue. RNA sequencing of adipose samples from 749 Icelanders shows that the transcripts generated from the variant allele in heterozygotes ($N = 26$) were wrongly spliced producing two new isoforms (Fig. 3). In one isoform, four extra base pairs are added to the end of exon 5, resulting in frameshift and introduction of a STOP codon after 93 amino acids (coded by exons 6 and 7). The second isoform is characterized by read-through into intron 5, with STOP occurring 35 amino acids into the intron. Thus, both isoforms are predicted to produce truncated proteins. Analysis of RNA expression levels demonstrated that heterozygous carriers have lower total *PRPH* transcript levels (Effect = −22.7% (95% CI = −33.0% to −10.9%), $P = 4.0 \times 10^{-4}$) than non-carriers ($N = 723$) (Fig. 4a, b). By comparing usage of the correct splice junction (SJ) at the variant site between non-carriers and heterozygotes, we see that carriers have half the usage of non-carriers of the correct SJ (Effect = −57.2% (95% CI = −68.0 to −42.9%) $P = 1.2 \times 10^{-8}$) (Fig. 4c). Furthermore, variant transcripts represent 35% of the total transcript fragments (total transcripts $N = 478$; 309 normally, and 169 incorrectly spliced variant transcripts), consistent with partial nonsense mediated decay of the variant transcript (Fig. 4b). As the variant is situated in an intron, correctly spliced RNA transcripts cannot be assigned to either allele. In an attempt to further assess whether the variant copy could produce a transcript with correct splicing, we looked for a marker in exon 5 or 6 that might help phasing the correctly spliced fragments. We found one, rs73112143, that tags the variant allele in three of the rs73112142-A carriers, and in all cases the alternate alleles were on distinct haplotypes. Utilizing this marker (rs73112143), we find no fragments supporting correct

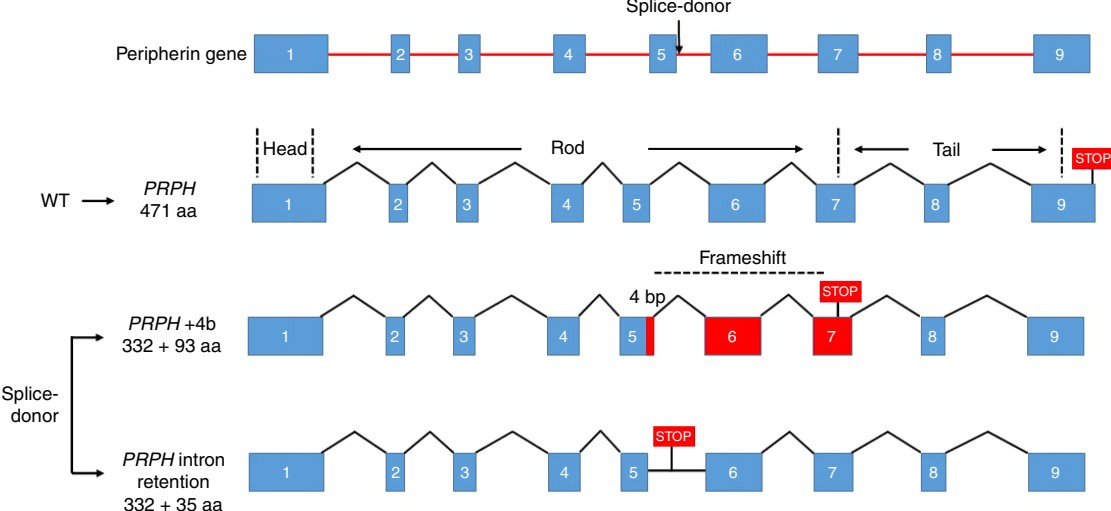

**Fig. 3** *PRPH* contains nine exons. The wild-type (WT) protein consists of a head domain expressed by exon 1, a rod domain expressed by exons 2–7, and a tail domain expressed by exons 7–9. The *PRPH* splice donor variant rs73112142-A is in the first base site after the fifth exon. Defective splicing generates two truncated frameshift variant proteins. In both cases the truncated proteins retain the first 332 amino acids; then, either four extra base pairs are added at the end of exon 5 putting protein out of frame after exon 5 and creating a STOP after 93 amino acids (coded by exons 6 and 7), or read-through into intron 5 occurs, with STOP after 35 amino acids (encoded by intron 5)

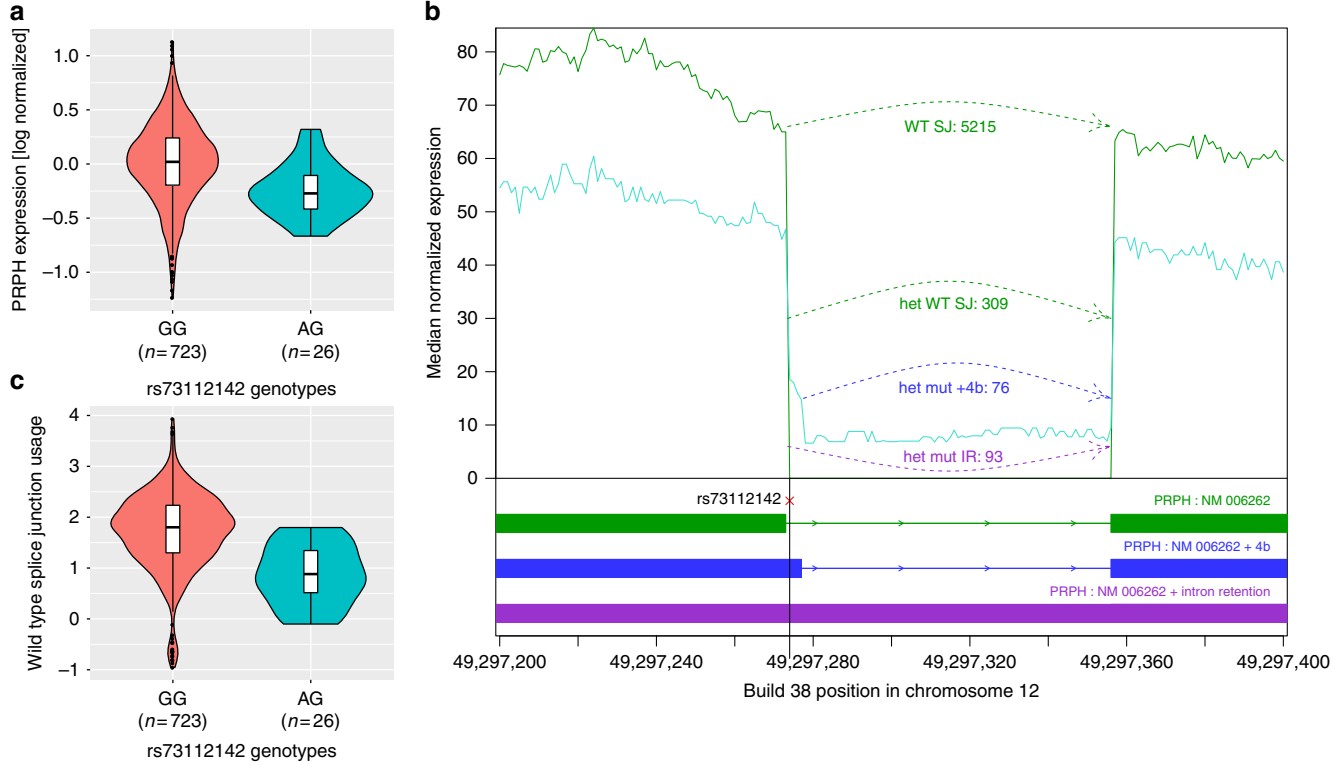

**Fig. 4** Expression levels of *PRPH* by genotype. **a** Expression levels of *PRPH* mRNA from adipose samples ($N = 749$). The violin plot represents density of expression estimates (log-normalized and adjusted for covariates), stratified by rs73112142 genotypes (Effect $= -22.7\%$ (95% CI: $-33.0$ to $-10.9\%$), $P = 4.0 \times 10^{-4}$). The bottom and top of each box represent the first and third quantiles, the line inside the box is the median, and whiskers represent the ±1.5 times the interquartile range. The filled circles correspond to expression values representing outliers that lie beyond the extremes of the whiskers. **b** RNA-seq coverage over the *PRPH* splice junction (SJ) site. The splice-donor variant, rs73112142, is on the black vertical line. *Y*-axis: median normalized expression in non-carriers (green, $N = 723$) compared to heterozygotes (turquoise, $N = 26$). Shown are fragment counts per allele (allele expression inferred): Normally spliced fragments in green for non-carriers (WT SJ; 5215 fragments) and heterozygous carriers (het WT SJ; 309 fragments), fragments with A-allele and 4 bp addition in blue (het mut + 4 bp; 76 fragments), fragments with A-allele and intron read-through in purple (het mut IR, 93 fragments). **c** *PRPH* wild-type splice junction (SJ) usage. *Y*-axis: SJ counts normalized for sequencing depth and log transformed. *X*-axis: Stratified by rs73112142 genotype, Effect $= -57.2\%$ (95% CI: $-68.0$ to $-42.9\%$), $P = 1.2 \times 10^{-8}$

splicing from the variant allele. While the number of fragments that could be assigned to the haplotype containing the mutated allele is small (19 normally spliced and 8 variant transcripts), the data are consistent with complete disruption of the splice site. In summary, the expression data indicate that no correctly spliced transcripts are generated from the variant allele and there is no compensation in expression from the major (WT) allele in heterozygous carriers.

**rs73112142-A precludes the formation of peripherin filaments**. To gain understanding of the effect of the two variant isoforms generated from rs73112142-A on the formation and function of peripherin, we overexpressed cDNAs for both isoforms as well as WT *PRPH* in SW13 cells, a human adrenal carcinoma cell line that lacks IF proteins. Western blot analysis of these cells detected the truncated *PRPH* proteins of the expected size (Supplementary Fig. 4). Homopolymeric IF networks, typical for class-III IFs, were formed in the majority of cells overexpressing WT peripherin (87.9% ± 2.3%) (Fig. 5a, d). Conversely, in SW13 cells overexpressing the 4bp-isoform or the intron-retention isoform, no filamentous structures were formed. Furthermore, cells transfected with the 4bp-isoform presented with small cytoplasmic puncti (47.3% ± 12.6%), singular large perinuclear or nuclear inclusions (32.9% ± 11.9%), or larger more fibrous bundles (19.9% ± 5.5%) (Fig. 5b, d). The intron-retention splice variant principally formed small inclusions (72.6% ± 12.3%) with

some cells forming large (11% ± 6.3%) and fibrous inclusions (16.5% ± 8.6%) (Fig. 5c, d). Taken together, peripherins produced by the two mutated isoforms do not form homopolymeric IF networks, indicating that rs73112142-A homozygotes lack functional peripherin.

**rs723112142-A does not confer risk of common neuropathies**. To evaluate whether the effects of the variant on SNAP predisposes to common neuropathies or neuropathic risk factors, we studied its association with the following neuropathies and neuropathy-related phenotypes defined by ICD-10 diagnostic codes; G59.0 diabetic neuropathy, G60 hereditary and idiopathic neuropathies, G61 inflammatory neuropathy, G62 other polyneuropathies, G56.0 carpal tunnel syndrome (an entrapment neuropathy), I10 hypertension, I95.1 orthostatic hypotension, E11 type-2 diabetes, and F10 alcohol dependence. Additionally, we studied the potential associations with blood factors associated with PN development, including vitamin $B_{12}$, fasting glucose, TSH, and various immunoglobulins linked to neuropathies in dysproteinemias; specifically IgM, IgG, and IgA[17], and with neuropathic pain as measured by the DN4 questionnaire[25]. As the splice-donor variant does not associate with any of these diagnoses or traits (all $P > 0.05$ in both additive and recessive models, Supplementary Table 3), the association of rs73112142-A with SNAP does not appear to mediate risk of these neuropathies or affect the neuropathy-associated traits.

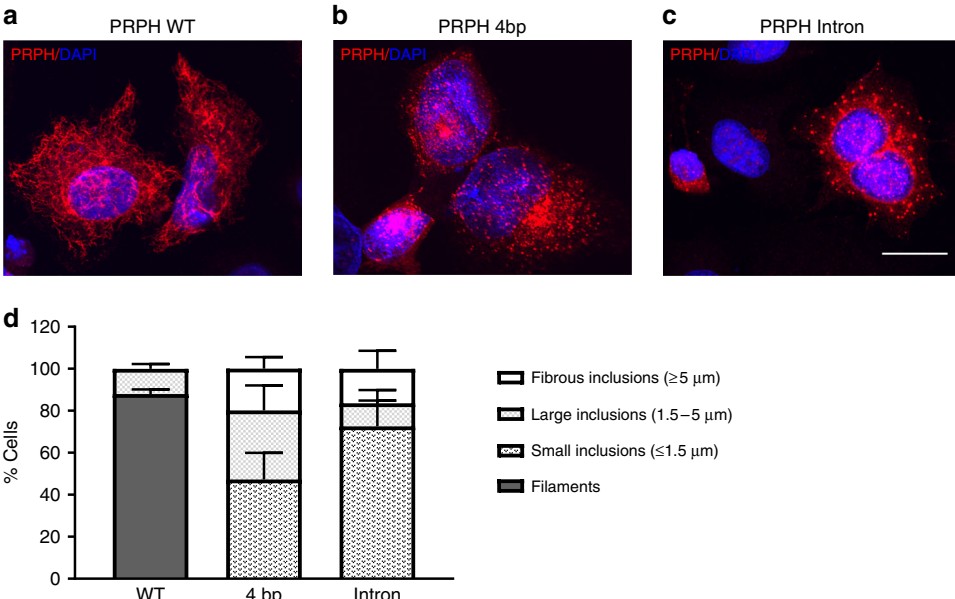

**Fig. 5** Truncated *PRPH* variants generate protein inclusions. SW13 cells were transfected with **a** wild-type (*PRPH* WT) using $n = 332$ cells, **b** 4bp-splice variant (*PRPH* 4 bp) using $n = 545$ cells, or **c** intron retention splice-variant (*PRPH* Intron), cultured for 48 h and labeled with an antibody raised against peripherin and counterstained with DAPI (4′,6-diamidino-2-phenylindole) for nuclear visualization, using $n = 177$ cells. Scale bar represents 10 μm. **d** Peripherin filaments and inclusions were quantified from the three independent experiments ($n = 177–545$). Inclusions were subdivided by size; fibrous inclusions ≥5 μm, large inclusions 1.5–5 μm, and small inclusions ≤1.5 μm ($n = 177–545$). Error bars represent standard deviations (SD)

We also tested the three sequence variants reported in the GWAS catalog (URLs) to associate with PN secondary to treatment with neurotoxic medications[7–9], and find that they do not associate with SNAP or SNCV ($P > 0.05$) (Supplementary Table 4).

Finally, although the evidence is weak, *PRPH* has been linked to amyotrophic lateral sclerosis (ALS) (OMIM #170710). Hence, we studied the association of rs73112142-A with ALS (ICD-10 diagnosis G12.2-Motor Neuron Disease) in Iceland ($N_{case/control} = 272/368,885$), finding no evidence of association (OR = 1.45, $P = 0.29$).

**rs73112142-A confers risk of a sensory-negative neuropathy.** To characterize phenotypic effects of the splice-donor variant, we re-contacted rs73112142-A carriers for an in-depth neurological assessment (Methods). Of the 43 carriers who responded to recall, 9 were homozygotes and 34 heterozygotes. We also recruited 26 non-carrier controls, sex- and age-matched to carriers. A neurologist and trained research nurses blinded to carrier status evaluated the recruited subjects (Methods). Given possible axonal pathology of the sural nerve as suggested by the association of rs73112142-A with SNAP, but not SNCV[1,2,4,11], standardized assessment scales were used to evaluate signs and symptoms of PN, including the Diabetic Neuropathy Symptom Score (DNS) that has a high predictive value in screening for diabetic neuropathy[26], the Toronto Clinical Neuropathy Score (TCNS) that is valid for a wide spectrum of polyneuropathies[27], and the Utah Early Neuropathy Scale (UENS) that is sensitive to small-fiber (unmyelinated) and early neuropathy detection[28].

All recalled individuals were generally healthy, reporting no diabetes, alcohol misuse, cancer history, treatment with cytotoxic medications, or neurological conditions affecting the central nervous system. All had normal blood levels of $B_{12}$, fasting glucose, and TSH (Supplementary Table 5). As a group, homozygotes have higher scores (more affected) than

heterozygotes and non-carriers, on all standardized PN assessment scales (Supplementary Table 5). As re-called subjects were without known PN risk factors, the scale most sensitive to early neuropathy detection (UENS[28]) was used to classify PN status. Five out of nine homozygotes had predominantly sensory polyneuropathy as determined by a score of 5 or more on the UENS[28]. Of these, only one had been diagnosed earlier with a PN of unknown etiology at the age of 40 according to interview. In comparison, out of 26 non-carriers, 2 had PN; an 8% prevalence consistent with population prevalence estimates for older individuals[3]. The primary symptom reported by affected homozygotes was distal numbness (toes and heals) and they recalled being significantly younger when first experiencing their PN symptoms (M = 20 years, range 10–35), than affected heterozygotes or non-carriers (M = 57 years, range 45–70), $P = 1.8 \times 10^{-4}$. While NC results are not part of the UENS criteria, NC testing of the sural nerve in the re-called sample confirm that homozygous carriers of the splice-donor variant have markedly lower SNAP (M = 4.0 μV, 95% CI = 2.9–5.2) compared to non-carriers (M = 14.5 μV, 95% CI = 10.7–18.4), while as previously observed in the discovery sample, SNCV results were comparable between genotype groups (Supplementary Table 5). Thus, standardized assessments of PN signs and symptoms show that the variant primarily affects small unmyelinated nerve fibers, however, also with larger and myelinated nerve fiber involvement, as indicated by less great-toe vibration detection time in homozygotes than in non-carriers, and to some extent distal motor fiber involvement, as indicated by decreased great-toe extension (Supplementary Table 5).

In summary, we observe an increased risk of an early-onset, mild and predominantly sensory peripheral polyneuropathy with axonal loss, among rs73112142-A carriers. Under an additive model, the association (OR = 4.5, (95% CI = 1.3–14.7), $P = 0.015$, $N = 69$) is driven primarily by homozygotes. Under a recessive model, the risk for homozygotes is 26-fold (OR = 26.1 (95% CI = 3.1–218.1), $P = 2.6 \times 10^{-3}$, $N = 9$).

## Discussion

In this first GWAS on sural NC, we find that a low-frequency, LoF splice-donor variant, rs73112142-A in *PRPH*, associates with markedly reduced SNAP but not with SNCV. *PRPH* encodes a class-III IF protein expressed in peripheral nerves that, along with other neurofilaments and microtubules, is involved in the development and maintenance of the axonal cytoskeleton[22–24].

Through RNA-sequence analysis of adipose samples from rs73112142-A heterozygous carriers, we show that the mutated allele generates two new RNA isoforms that are predicted to produce C-terminal truncated peripherins. Neither isoform has an intact rod domain which is necessary for IF assembly. Indeed, the variant proteins are unable to form homopolymeric networks when over-expressed in vitro. Hence, they are also unlikely to heteropolymerize with other neurofilaments, with which periph-erin is co-expressed, such as the neurofilament triplet proteins (neurofilament light (NFL), medium (NFM), and heavy (NFH) chain)[22,23]. Studies by others have shown that defective splicing in *PRPH* generates aggregation-prone C-terminal truncated per-ipherin, and that such a protein acts in a dominant-negative manner, as expression leads to an almost total collapse in another class-III IF (Vimentin) network[29]. However, our data argue against a dominant-negative effect of rs73112142-A, since it works in an additive manner evidenced by a clear dose-dependent genotype effect on SNAP (Table 2). If dominant negative, the effect on homozygotes would be similar to that of heterozygotes.

In addition to detecting the two new *PRPH* RNA isoforms generating truncated proteins, the RNA sequencing also demonstrates that the levels of peripherin in rs731121142-A carriers are less than in non-carriers. Since the reduction in generation of full-length protein, in both heterozygotes and homozygotes, is accompanied by the presence of truncated pro-teins, we cannot state if the association of the variant allele with reduction in SNAP is because of loss of full-length protein, pre-sence of truncated protein, or a combination of the two.

Developing neurons sequentially express distinct IF proteins at different stages of cell differentiation correlating with altered morphologies during neuronal development[23]. *PRPH* expression is greatest prenatally, during the axonal growth phase, and decreases postnatally, consistent with a role in axonal guidance during development and neurite elongation[23]. It is also known to play a role in axonal transport[30,31] and in repair after axonal damage[24,32]. Interestingly, emerging evidence suggests that dis-ruption in axonal transport is one of the molecular mechanisms affected in HSAN[5]. Mutations in at least 70 genes have been linked to the pathogenesis of HSAN and CMT[4–6], but *PRPH* is not among them.

In terms of clinical characterization of the effects of the *PRPH* splice-donor variant, we did not find associations with relevant neurological or neuropathic phenotypes available to us. However, with clinical follow-up of carriers, including standardized PN assessment, we do observe pathological alterations associated with the variant, in that we confirm that homozygous carriers not only have markedly reduced SNAP from a young age, they are also at risk of an early-onset, axonal PN, presenting with mild, pre-dominantly sensory-negative symptoms.

Peripheral nerves are composed of many nerve fibers of dif-ferent diameters, degrees of myelination, and hence, different conduction properties[1,2,11,15]. While changes in conduction pat-terns of peripheral nerves can indicate whether the underlying pathology is due to axonal loss, myelin loss, or both, conventional NC studies measure only the 20% of the fastest conducting sen-sory fibers with the largest diameter[11]. Small fiber neuropathies, affecting the other 80% of fibers generally not detected by NC studies, usually are characterized by prominent symptoms of pain or numbness and conventional sensory studies may be normal[11].

In light of the markedly lowered SNAP (but not SNCV) and sensory-negative PN with onset as early as childhood, we propose that the phenotype observed in carriers of the *PRPH* splice-donor variant may be due to a developmental disorder involving axonal loss of primarily unmyelinated fibers of the sural nerve. Indeed, studies of *PRPH* knockout (−/−) mice show that they have substantially fewer (34%) L5 unmyelinated sensory fibers extending from the dorsal root ganglion, while myelinated L5 neurons are unaffected[33]. However, lacking peripherin does not result in other neurological phenotypes in mice and they develop and reproduce normally[33]. Although large myelinated fibers are also affected in human rs73112142-A carriers (as observed by decreased sense of great-toe vibration and large fiber sensation, Supplementary Table 5), the *PRPH* (−/−) mouse phenotype is consistent with results of our clinical follow-up study. Human homozygous carriers of the LoF *PRPH* variant were individuals in relatively good health, apart from their significantly reduced SNAP and increased risk of the mild, sensory-negative PN.

Finally, through measuring sural NC in a large population-based sample of Icelanders, we had the opportunity to study correlations of SNAP, SNCV, demographics, and anthropo-metrics in a much larger sample size than used in previous normative studies[15,16]. Our analysis shows the importance of adjusting sural NC measures for both clinical use and research. The observation that both SNAP and SNCV require adjustments within sex is novel and has not been clearly shown before[15,16].

The path from GWAS of complex traits to their underlying biology can be tortuous. Here, by performing GWAS on traits defined by measures of physiological function of a peripheral nerve, we have discovered one such path. We have connected a gene without previous phenotype associations to functionally confirmed biological underpinnings of an axonal, sensory-negative polyneuropathy. Further studies of the clinical impact and prognosis of this novel genetic form of PN are required.

## Methods

**Subjects, phenotypes, and whole-genome sequencing in Iceland**. By partici-pating in various deCODE study projects over the past 20 years, over 150,000 Icelanders have been genotyped using Illumina SNP arrays and expectation values for genotypes for their un-genotyped relatives can be calculated based on genea-logical relationships. The process used to whole-genome sequence the Icelandic population and the imputation approaches used in this study have been described in detail[34,35]. To summarize, we sequenced the whole genomes of 28,075 Icelanders using Illumina technology to a mean depth of at least 10× (median 32×). SNPs and indels were identified and their genotypes called using joint calling with Graphtyper[36]. Genotype calls were improved by using information about haplotype sharing, taking advantage of the fact that all the sequenced individuals had also been chip-typed and long range phased. The 37.6 million variants passed the high-quality threshold were then imputed into 155,250 Icelanders who had been genotyped with various Illumina SNP chips and their genotypes phased using long-range phasing[18]. Using genealogic information, the sequence variants were imputed into relatives of the chip-typed to further increase the sample size for association analysis and increased the power to detect associations. All of the variants that were tested had imputation information over 0.8.

All data were collected through studies approved by the National Bioethics Committee and the Data Protection Authority in Iceland. The Icelandic sample with NC measures consists of individuals recruited: (a) for the deCODE Health Study (protocol number: VSNb2015120006/03.01 with amendments) and (b) for a study on the genetics of chronic and neuropathic pain (protocol number: VSNb2012090009/03.12 with amendments). As part of both studies, participants answer comprehensive health and lifestyle questionnaires including the DN4 (ref. [25]) to screen for neuropathic pain, and undergo several physiological tests, including sural NC testing[16]. Participation also includes blood sample donation and permission to access diagnostic data from hospital records. All participants signed informed consent before blood samples were drawn. All personal identifiers were encrypted by a third-party system overseen by the Icelandic Data Protection Authority[37].

NC was measured on the sural nerve in the right lower calf region (left if right leg was injured) that transmits signals from the posterior lateral corner of the leg and lateral foot and fifth toe to the spinal cord at S1 and S2. NC was measured using DPNCheck® (Neurometrix®, https://www.dpncheck.com), a handheld device that measures SNCV and SNAP with a disposable biosensor at a fixed distance of

9.22 cm from the electric stimulating probe[16]. The stimulating probe is placed over the sural sensory nerve on the posterior side of the lateral malleolus of subjects. The sural nerve is orthodromically stimulated with max 420 μV 4–16 times within 10–20 s. Key outcome measures are sensory latency, which correlates directly with SNCV and peak-to-peak amplitude (SNAP)[11,16]. Research nurses obtained NC measures in our recruitment facility, after participants had spent over 30 min in room temperature of at least 25 °C, ensuring stable skin temperature of participants' legs. The device corrects NC measures for skin temperature between 23 and 28 °C and prevents testing when skin temperature is below 23 °C resulting in missing data[16]. If SNAP is <2 μV but detectable, results are adjusted to 0 by the device[16]. Normative data are provided by the manufacturer based on sural nerve measures from a total of 527 healthy subjects[16]. Results suggested corrections for age as SNAP correlates with age, declining about 1 μV for every decade[16]. SNCV was reportedly dependent on both age and height, decreasing by 1.3 m s$^{-1}$ for every decade and 2.0 m s$^{-1}$ for every 10 cm of height[16]. Normal values for clinical screening purposes recommended by this study were SNAP > 4 μV and SNCV > 40 m s$^{-1}$ (ref. [16]) in agreement with sural NC reference values reported by the Normative Data Task Force (NDTF) of Electrodiagnostic Medicine[15].

Carriers of rs73112142-A and age- and sex-matched non-carriers were re-called for neurological assessment according to a study protocol approved by the National Bioethics Committee (#VSNb2014010013/03.12). Subjects and recruitment staff were blind to carrier status. Subjects were evaluated by a neurologist and trained research nurses, also blinded to carrier status, using clinical examination scales widely used in the assessment of PN. These included the DNS[26], modified TCNS that is valid for a wide spectrum of polyneuropathies[27], the UENS that is most sensitive to small-fiber (unmyelinated) neuropathy detection[28], and the Romberg balance and co-ordination test (http://www.pitt.edu/~whitney/sls.htm). Neurologist-trained research nurses conducted neuropathy- and pain history interviews, PN-evaluation according to protocols and performed quantitative sensory testing (QST). QST measures were performed on both lower and upper limbs. In lower limbs, heat and cold detection and heat and cold pain were measured in the dermatome of the sural nerve, using the Medoc Pathway Pain and Sensory Evaluation System (https://medoc-web.com/products/pathway/).

**Association analyses.** Our initial dataset consisted of 7680 individuals with nerve conduction measurements. Of these, we had height measurements for 7663 and leg fat mass (DEXA) measurements for 7485. Of individuals with NC and height measurements ($N = 7663$), 670 had missing SNCV of which 604 also have missing SNAP. Of individuals with NC, height, and leg fat mass measurements ($N = 7485$), 592 had missing SNAP of which 591 also had missing SNVC. In the end, we tested variants for association with SNCV in a dataset of 6979 individuals and SNAP in a dataset of 6879 individuals, a total sample of 7045 individuals. Based on our correlation observations, the quantitative traits SNAP and SNCV were rank-based inverse normal transformed separately for each sex. SNVC was adjusted for age at measurement and height while SNAP was adjusted for age at measurement, height, and leg fat mass, using a generalized additive model[38]. For each sequence variant, a linear regression model, using the genotype as an additive covariate, the transformed quantitative trait as a response and assuming the variance-covariance matrix to be proportional to the kinship matrix, was used to test for association. We used LD score regression to account for distribution inflation in the dataset due to cryptic relatedness and population stratification[39]. With a set of 1.1 million variants, we regressed the $\chi^2$ statistics from our GWASs against LD score and used the intercepts as a correction factors. To examine if effects differed by sex, we assessed the effects for males and females separately and tested them for statistical heterogeneity with a Q-test[19].

Logistic regression was used to test for association between sequence variants and binary traits. Other available individual characteristics that correlate with disease status were also included in the model such as sex, county of birth, current age or age at death (first and second order terms included), blood sample availability for the individual, and an indicator function for the overlap of the lifetime of the individual with the time span of phenotype collection.

**Significance thresholds.** The thresholds for genome-wide significance were estimated from the Icelandic data and corrected for multiple testing with a weighted Bonferroni adjustment using as weights the enrichment of variant classes with predicted functional impact among association signals[40]. With 37.6 million sequence variants in the Icelandic data the weights given in Sveinbjornsson et al.[40] were rescaled to control the family-wise error rate. This resulted in significance thresholds of $2.5 \times 10^{-7}$ for loss-of-function variants, $5.0 \times 10^{-8}$ for moderate-impact variants, $4.5 \times 10^{-9}$ for low-impact variants, $2.3 \times 10^{-9}$ for other variants within DHS sites, and $7.5 \times 10^{-10}$ for remaining variants.

**RNA sequencing analysis.** RNA sequencing analysis was performed on sub-cutaneous adipose tissue samples obtained from 749 Icelanders. RNA was isolated using Qiagen RNA midi kit according to the manufacturer's instructions. Concentration and quality of the RNA was determined with an Agilent 2100 Bioanalyzer (Agilent Technologies). RNA was prepared and sequenced on the Illumina HiSeq 2500 System according to the manufacturer's recommendation and aligned to the reference genome using the STAR software package. Gene expression

was computed based on personalized transcript abundances estimated using kallisto[41]. Association between variant and gene expression was estimated using a generalized linear regression, assuming additive genetic effect and log-transformed gene expression estimates, adjusting for measurements of sequencing artefacts, demography variables, and hidden covariates[42]. Uniquely aligning fragments overlapping SJs in *PRPH*, covering at least two base pairs on both sides, were counted for each sample. SJ usage was estimated for each sample, normalized by sequencing depth.

**Generation of *PRPH* splice variant plasmids.** The full-length PRPH cDNA (NM_006262.3) in pCMV6-Entry mammalian expression vector including a Myc-DDK tag was obtained from Origene (RC207561). Plasmids for the 4 bp- and intron-retention splice variants were generated by mutating the full-length PRPH plasmid in two separated PCR reactions using Q5 site-directed mutagenesis kit (New England BioLabs, E0554S). The 4bp-splice variant was generated by adding the first four bases of intron 5 (ATGA) with the mutagenesis primers F-5′ATGA AACGAGGCGCTGCTCAGGCAG-3′ and R-5′CGTGCCGCGCAGCCCGTC-3′, resulting in a plasmid with a premature stop codon, thus lacking expression of the Myc-DDK tag. To include expression of the Myc-DDK tag in the 4 bp splice variant plasmid, the premature stop codon and the cDNA sequence up until the Myc-DDK tag was deleted from the synthesized 4 bp splice variant plasmid with the mutagenesis primers F-5′-ACGCGTACGCGGCCGCTC-3′ and R-5′-GGCAC AGTCGTCTTTATATTTAAGGAGGCAAAAGAATGG-3′. The intron-retention splice variant plasmid was generated by adding the whole genomic sequence of intron 5 with the mutagenesis primers F-5′-GAAATGTTCTGCAACTGGCCCCT TCCACTCTCCTACCCCAGAACGAGGCGCTGCTCAGGCAG-3′ and R-5′-AT CGTCCGCTCCCCGGGCCCGAGCGCGCAGCTTCGTACTCATCGTGCCGCG CAGCCCGTC-3′, resulting in a plasmid with a premature stop codon lacking expression of the Myc-DDK tag. The premature stop codon and the cDNA sequence up until the Myc-DDK tag was deleted from the synthesized intron-retention splice variant plasmid with the mutagenesis primers F-5′-ACGCG-TACGCGGCCGCTC-3′ and R-5′-ACTGCCTGAGCAGCGCCTC-3′ to include expression of the Myc-DDK tag. Mutated plasmids were transformed into NEB Stable Competent *E. coli* (New England BioLabs, C3040H) and plated on LB agar plates containing 25 μg ml$^{-1}$ kanamycin. Colonies were expanded in LB medium containing 25 μg ml$^{-1}$ kanamycin. Plasmids were purified using Qiagen plasmid maxi kit (Qiagen, 12163), following the manufacturer's protocol. Sequences of WT-pCMV6-Entry, 4bp-pCMV6-Entry and intron-retention-pCMV6-Entry plasmids were confirmed by Sanger sequencing.

**Western blot analysis.** Cell pellets were lysed in 100 μL RIPA buffer (150 mM NaCl, 1% TritonX, 0.5% sodiumdeoxycholate, 0.1% SDS, 50 mM Tris pH8) with Halt Protease and phosphatase inhibitor (ThermoFisher, #78445) and incubated for 10 min on ice. Samples were spun at $14,000 \times g$ for 15 min at 4 °C. The supernatant was transferred to a fresh tube and the protein content of lysates was determined by BCA assay (ThermoFisher, #23227). Samples were mixed with NuPAGE LDS sample buffer (ThermoFisher, #B0007) and reducing agent (ThermoFisher, #B0009), heated for 10 min at 70 °C, and loaded on a 4–12% BT bolt plus gel (ThermoFisher, #NW04120BOX). The gel was equilibrated for 5 min in 20% EtOH and transferred onto a PVDF membrane (ThermoFisher, # IB24002) with iBlot 2 (ThermoFisher) running the P0 program. Following the transfer, membranes were dried for 1 h. Membranes were activated in MeOH for 20 s, washed with water and TBS, and subsequently blocked in TBS 5% milk for 1 h. Blots were incubated with primary antibodies overnight at 4 °C in TBS 5% milk + 0.1% Tween20. Antibodies used were anti-DDK at 1:5000 (Origene, TA50011-1) and anti-mouse IgG (H + L) IRDye 800CW at 1:10,000 (LI-COR, #926-32212). Membranes were scanned with Odyssey CLx and subsequent analysis performed in the image studio software (LI-COR).

**Cell culture, transient transfection, and immunofluorescence.** The SW13 adrenocortical carcinoma cell line was purchased from ATCC (CCL-105) and cultured at 37 °C, 5% CO$_2$ in Dulbecco's modified Eagle's medium (31966-021; Gibco) supplemented with glutamax, 10% fetal bovine serum (10500-064; Gibco), 100 U ml$^{-1}$ penicillin and 100 μg ml$^{-1}$ streptomycin (15140-122; Gibco). Cells were grown on poly-L-lysine (P4707; Sigma Aldrich) coated glass coverslips (P231.1; Carl Roth GmbH Co.) and transiently transfected for 48 h using Fugene (E2312; Promega) according to the manufacturer's instructions. For immuno-fluorescent staining, cells were washed with DPBS (14190-144; Gibco) and fixed with MeOH for 5 min at −20 °C, washed with DPBS, and incubated for 1 h at RT with an anti-peripherin antibody (1:1000, AB1530; Millipore) in blocking buffer (DPBS with 5% bovine serum albumin (BSA, 15260-037; Gibco) and 0.3% Triton-x (708007; Merck). Coverslips were washed with DPBST (0.05% tween (P1379, Sigma)) and incubated for 1 h at RT with goat anti-rabbit IgG AF546 (A11035; Thermo Fisher Scientific) secondary antibody in DPBST with 0.5% BSA. Nuclei were stained with DAPI (D9542; Sigma Aldrich). Coverslips were subsequently mounted on glass slides using ProLong Diamond Antifade Mountant (P36965; Life Technologies) and sealed with nail polish. Fluorescent imaging was carried out on a FV1200 Olympus inverted confocal microscope (Olympus) and image processing in Fiji[43]. For quantification, cells with small inclusions only were categorized as

having small inclusions; cells with large or fibrous inclusions were categorized with large or fibrous inclusions regardless of the presence of small inclusions No observable difference was found to be between the *PRPH* protein expression with or without Myc-FLAG tag. A list of all primers used in the study and their sequences are available in Supplementary Table 6.

**URLs**. GTEX data (http://www.gtexportal.org) were accessed in September 2018. GWAS catalog (https://www.ebi.ac.uk/gwas/home) was accessed in September 2018. Human Protein Atlas (https://www.proteinatlas.org/) was accessed in September 2018. LD Score Database (ftp://atguftp.mgh.harvard.edu/brendan/1k_eur_r2_hm3snps_se_weights.RDS) was accessed in September 2018. UCSC Genome Browser (https://genome.ucsc.edu/) was accessed in September 2018.

**Reporting summary**. Further information on experimental design is available in the Nature Research Reporting Summary linked to this article.

## Data availability

The sequence variants from the Icelandic population whole-genome sequence data have been deposited at the European Variant Archive under accession code PRJEB15197. The authors declare that the data supporting the findings of this study are available within the article, its Supplementary Information file, and upon request. The GWAS summary statistics for SNAP and SNCV are available at https://www.decode.com/summarydata. The source data underlying Figs. 1–5 are provided as a Source Data file.

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

## Acknowledgements

We thank all participants in deCODE studies for their valuable contribution to research, especially the participants of the deCODE Health Study and the deCODE Study on Genetics of Chronic and Neuropathic Pain. We also thank the research staff at the Patient Recruitment Center, and all colleagues who contributed to phenotype ascertainment, recruitment, collection of data, sample handling, and genotyping. The financial support from the European Commission to the NeuroPain project (FP7#HEALTH-2013-602891-2) and the National Institutes of Health (R01DE022905) is acknowledged.

## Author contributions

G.B., E.V.I., K.B., S.B., S.S.G., G.H.H., O.T.M., G.M., P.M., D.F.G., H.H., H.S., U.T., G.L.N., H.H., P.S., T.E.T. and K.S. designed the study and interpreted the results. G.B., E.V.I., K.B., S.B., S.S.G., G.A.A., G.H.H., A.H., Adalbjorg Jonasdottir, Aslaug Jonasdottir, I.J., A.M.K., O.T.M., G.M., P.M., T.R., A. Sigurdsson, G.S., A. Skuladottir, V.S., U.S., G. Thorgeirsson, G. Thorleifsson, D.F.G., H.H., H.S., U.T., G.L.N., P.S., T.E.T. and K.S. analyzed the data. G.B., S.S.G., R.B., A.H., T.R., G.S., A. Skuladottir., V.S., U.S., G. Thorgeirsson, A.V., H.H. and T.E.T. performed recruitment and phenotyping. G.B., E.V.I., K.B., S.B., S.S.G., G.H.H., A.H., P.M., G. Thorgeirsson, D.F.G., H.H., H.S., U.T., G.L.N., P.S., T.E.T. and K.S. drafted the manuscript with input and comments from all other authors.

## Additional information

**Competing interests:** G.B., E.V.I., K.B., S.B., G.A.A., G.H.H., A.H., Adalbjorg Jonasdottir, Aslaug Jonasdottir, I.J., A.M.K., O.T.M., G.M., P.M., T.R., A. Sigurdsson, G.S., A. Skuladottir, V.S., U.S., G. Thorgeirsson, G. Thorleifsson, D.F.G., H.H., H.S., U.T., G.L.N., P.S., T.E.T. and K.S. are employees of deCODE genetics/Amgen, Inc. The remaining authors declare no competing interests.

