## [Peer Review File · Nature Communications]

Reviewer #1 (Remarks to the Author):

This is an excellent report of a well-performed novel genetic study of two nerve conduction phenotypes (SNAP and SNCV).

The study has many strengths, primarily in its efforts to perform i) a large well-powered genome-wide association study (GWAS), ii) follow-up functional studies, and iii) clinical follow-up.

In my view, there are only a few limitations, which although relatively minor, addressing them would improve the papers readability and interpretation:

First and foremost, given the significant difference in SNAP measures between males and females, many readers (incl. myself) would like to see results from sex-specific (GWAS) analyses to allow further interpretation, including homogeneity of effect on risk of the associated splice donor variant (rs73112142-A).

Secondly, given the GWAS was performed in a single, albeit well-powered sample, replication is lacking. The authors should at least acknowledge/discuss this limitation.

Third, given the low frequency of the implicated variant, I would like the authors to discuss whether there likely exists additional rare and/or common variants associated with SNAP (and SNCV). For example, did the LD score analyses provide evidence for SNP-based heritability?

Other more minor requests include the following: 1) please list and cite the specific computer software program/package used to perform the association analyses; 2) make clearer the multiple tests [e.g., phenotypes? genetic models?] being corrected for in the 'Significance thresholds' methods [i.e., the listed thresholds are correcting for more than just the maximum of 37.6 million variants tested]; 3) Replace computer number notations to scientific notation of p-values listed in the Figure 2 legend; and 4) use consistent naming in the Author contributions re GThorleifsson, GThorg, GThorl.

Reviewer #2 (Remarks to the Author):

The manuscript by Bjornsdottir et al associates a splice-donor variant in the PRPH gene to be associated with a nerve conduction phenotype with a suggestive link to peripheral neuropathy.

The manuscript is a great example of using a large scale “genetics first” strategy to improve our understanding of the genetic basis of human physiology.

The manuscript is clearly written and the experimentation is convincing. I have just a few points:

- 1) The association of the PRPH splice variant to SNAP is convincing. The link between SNAP and neuropathy might need more clarification. After all, as this reviewer understands it, the diagnostic criteria of peripheral neuropathy do not include SNAP. This should be clarified for an audience which does not consist of neurophysiologists. In other words what is the role of SNAP in peripheral neuropathy.
- 2) Related to point (1), on lines 116-117 the sentence is written in a way that the reader understands that a PN diagnosis is met if SNAP and SNCV are below reference values. Is that really so? Isn't PN after all a clinical diagnosis?
- 3) The study using PRPH minor allele homozygotes is a brilliant way to understand the consequences of the variant and having an infrastructure where such cases can be recalled is unique. However, understandably, the sample is small. Setting the diagnosis of PN remains slightly unclear to the reader. On line 218 it is stated that symptoms are mild. This raises the question, do the findings reach the diagnosis of PN or should they be called “mild PN like symptoms”. Please clarify. This is especially unclear as the authors stress on the younger age of onset of PN symptoms. If the symptoms are very mild, what is the specifics of evaluating the age of onset.
- 4) On line 222 there is a reference to Supplemental Table 6. At least this reviewer cannot find Suppl Table 6.
- 5) It would be helpful if the authors would report what they observe in UKBB. Is the variant imputed in the UKBB data. Moreover, it would be good to discuss the possibilities for replicating the presented results. Was there an attempt for this?
- 6) On line 223 the authors state: “..we observe an increased risk of early-onset, predominantly sensory peripheral polyneuropathy”. The p-value is nominally significant, so maybe the “topic sentence” could be somewhat tuned down. Is the p-value corrected for multiple testing?

Reviewer #3 (Remarks to the Author):

In this paper, the authors have performed a GWAS of sural nerve conduction amplitude (SNAP) and velocity (SNCV) in 7,045 Icelanders. They found that a splice donor variant in the gene encoding the intermediate filament protein, peripherin associates with decreased SNAP, but not SNCV. Defective splicing due to this variant yields two truncated frameshift peripherin proteins, with a truncated rod domain as a result. Overexpression of the splice variant's isoform in a cell line devoid of

intermediate filaments showed that the variants do not allow formation of the normal filamentous structure of peripherin.

Overall, this is an interesting study that implicates peripherin dysfunction in sural nerve conduction amplitude, which has previously not been observed.

I have a few comments and/or questions:

1. The most important one has to do with the mechanism by which this variance would cause problems. The results of the transfection studies suggest that the splice variant isoforms cannot form filaments on their own, but the more relevant question is whether this results in a dominant negative failure of any intermediate filaments to form. This could easily be studied by co-transfecting wild-type peripherin with the variants.

2. A second possibility is suggested by the study of peripherin expression in adipose tissues, which showed that transcripts from the mutated allele were 41% fewer than those from the wild-type allele. This suggests that the results could be because of lower levels of peripherin. Can the authors comment on this.

3. The results mentioned in item 2, could be better described and explained. Since it is only shown in the supplementary figures (why?), the figure legend lacks sufficient detail and the experimental details are also not described. The fact that peripherin is expressed in adipose tissues is also a bit of a surprise to me. What levels are they expressed at, and in what cells? Peripherin is thought to be primarily in neurons.

4. Minor quibble: the SW13 Vim- cells are said to be devoid of filaments. To be precise, they are devoid of intermediate filaments, there are multiple other filaments (microfilaments/microtubules) present in these cells.

1. Reviewer #1 (Remarks to the Author):

This is an excellent report of a well-performed novel genetic study of two nerve conduction phenotypes (SNAP and SNCV). The study has many strengths, primarily in its efforts to perform i) a large well-powered genome-wide association study (GWAS), ii) follow-up functional studies, and iii) clinical follow-up. In my view, there are only a few limitations, which although relatively minor, addressing them would improve the papers readability and interpretation:

- 1.1. First and foremost, given the significant difference in SNAP measures between males and females, many readers (incl. myself) would like to see results from sex-specific (GWAS) analyses to allow further interpretation, including homogeneity of effect on risk of the associated splice donor variant (rs73112142-A).*

Response (1.1.): We agree that sex-specific results for the splice donor variant on SNAP were missing from the manuscript. They indeed show no significant heterogeneity of effect of the variant on SNAP between males and females, and have now been added to the revised manuscript:

- a) to results (page 5, lines 116-118): "The variant has comparable effects on SNAP in males ($\beta = -0.48$ SD, $P = 8.0 \times 10^{-7}$) and females ($\beta = -0.51$ SD, $P = 1.5 \times 10^{-6}$) (Heterogeneity $P = 0.83$)", and
- b) to table 2 (page 33), now showing sex-specific results for additive and recessive models, in both SNAP units (SD and $\log(\mu V)$), and heterogeneity P -values for the Q-test of difference in effects between males and females within each genetic model (additive and recessive).
- c) to methods (page 16, lines 384-385), with additional reference (nr. 37) for the Q-test of heterogeneity (Higgins et al., 2002).

- 1.2. Secondly, given the GWAS was performed in a single, albeit well-powered sample, replication is lacking. The authors should at least acknowledge/discuss this limitation.*

Response (1.2.): We agree that replicating our findings in a sample from another population would have strengthened our study. We widely sought, but could not find adequately powered samples in other populations with nerve conduction measures. Given the low minor allele frequency (MAF) in other European countries (0.73%, as stated in line 125), and even lower outside of Europe (EXAC, URLs), large and comparably phenotyped samples were required. The comprehensive phenotyping performed by the UK Biobank (where rs73112142-A MAF = 0.47%) does not include nerve conduction measures.

However, the association of the splice-donor variant in *PRPH* is several orders of magnitude more significant than our set *P*-value cutoff ($P = 5.8 \times 10^{-7}$ for loss-of-function variants, see Sveinbjornsson, G. et al.(2006). *Weighting sequence variants based on their annotation increases power of whole-genome association studies. Nat Genet 48, 314-7, PMID: 26854916*). Additionally, we provide evidence of the variant's effects in terms of both functional and clinical follow-up studies.

In response to the reviewer's comment and to clarify the lack of replication data to readers, we have added the following statement to the results section (starting in line 126, page 5): "We did not find adequately powered and comparably phenotyped samples from other populations, in which to replicate our findings."

1.3. Third, given the low frequency of the implicated variant, I would like the authors to discuss whether there likely exists additional rare and/or common variants associated with SNAP (and SNCV). For example, did the LD score analyses provide evidence for SNP-based heritability?

Response (1.3.): The reviewer is correct in that there possibly exist additional variants associated with SNAP and SNCV, especially in other populations. In our Icelandic data, where we have high quality sequencing and imputation of variants down to a frequency of 0.01% (see also response 1.4.1. below), we do not find other variants than the splice-donor variant rs73112142-A, significantly associating with SNAP or SNCV. This is clearly stated in the results section of the manuscript (page 5, lines 121-123), and demonstrated in the Manhattan plots for the SNAP and SNCV GWAS results in Supplementary Figure 2.

In terms of the reviewer's question on SNP-based heritability for SNAP or SNCV, we did not calculate this as our data do not allow for unbiased estimation of heritability. Our sample size is smaller than recommended for such calculations (see Yang et al. *Concepts, estimation and interpretation of SNP-based heritability. Nat Genet. 2017 Aug 30;49(9):1304-1310, PMID: 28854176*), and we only have one genome-wide significant finding. It has been the general experience that the phenotypic variance explained by association results tends to be small in such situations, and SNP-based heritability estimates are also expected to be unreliable (Yang et al. *PMID: 28854176*). Hence, it is our opinion that SNP-heritability calculation using our results would be unreliable (and small) due to the limitations of our sample, precluding meaningful general conclusions relevant to the underlying genetics of the traits in question.

1.4. Other more minor requests include the following:

1.4.1 please list and cite the specific computer software program/package used to perform the association analyses;

Response (1.4.1.): The software packages used for the association testing were developed in-house at deCODE. We describe them in our methods section, providing relevant references to further methodological details where required. However, in response to the reviewer's comment, and to

clarify our association analyses methods further, we have now added the following details to the methods section in the revised manuscript (page 13, lines 316-325):

“To summarize, we sequenced the whole genomes of 28,075 Icelanders using Illumina technology to a mean depth of at least 10X (median 32X). SNPs and indels were identified and their genotypes called using joint calling with the Genome Analysis Toolkit HaplotypeCaller (GATK version 3.4.07). Genotype calls were improved by using information about haplotype sharing, taking advantage of the fact that all the sequenced individuals had also been chip-typed and long range phased. The 37.6 million variants passed the high quality threshold were then imputed into 155,250 Icelanders who had been genotyped with various Illumina SNP chips and their genotypes phased using long-range phasing. Using genealogic information, the sequence variants were imputed into relatives of the chip-typed to further increase the sample size for association analysis and increased the power to detect associations. All of the variants that were tested had imputation information over 0.8.”

1.4.2. *make clearer the multiple tests [e.g., phenotypes? genetic models?] being corrected for in the 'Significance thresholds' methods [i.e., the listed thresholds are correcting for more than just the maximum of 37.6 million variants tested];*

Response (1.4.2.): In the method's "Significance thresholds" section referred to by the reviewer (page 16, lines 392-399), we describe how in our GWAS discovery studies, we correct for number of variants tested using the method referred to in Sveinbjornsson *et al.* (PMID: 2685491). The method uses as weights the enrichment of sequence annotations among association signals, resulting in better control for family-wise error rate (FWER) (*i.e.* false discovery rate in sets of items of inference in an analysis), than the Bonferroni method. Accordingly, we set the lowest significance *P*-value for common low-impact variants at 7.5×10^{-10} , an order of magnitude lower than a Bonferroni corrected *P*-value for 37.6 million variants tested ($0.05/37,600,000$ or 1.33×10^{-9}). In contrast, for variants annotated as having higher impact, the threshold is set at higher *P*-values. Thus, for the highest impact variants (LoF) such as the *PRPH* variant, the so-weighted threshold is 5.8×10^{-7} . Given that the observed *P*-value is 4 orders of magnitude lower than the appropriate threshold, the finding is clearly significant.

1.4.3. *Replace computer number notations to scientific notation of p-values listed in the Figure 2 legend; and*

Response (1.4.3.): This has been corrected.

1.4.4. *use consistent naming in the Author contributions re GThorleifsson, GThorg, GThorl.*

Response (1.4.4.): As pointed out by Reviewer #1, Gudmundur Thorleifsson was identified erroneously by two different abbreviations. This has been corrected. To explain why simple initials do not suffice is that 2 authors (Gudmundur Thorgeirsson and Gudmundur Thorleifsson), have the same initials (GT), same first name and same first 4 letters of their surnames (Thor), thus, requiring additional letters to identify them correctly (GThorg & GThorl). If the editors prefer another format (*e.g.* enumeration), please advise.

Reviewer #2 (Remarks to the Author):

The manuscript by Bjornsdottir et al associates a splice-donor variant in the PRPH gene to be associated with a nerve conduction phenotype with a suggestive link to peripheral neuropathy. The manuscript is a great example of using a large scale “genetics first” strategy to improve our understanding of the genetic basis of human physiology. The manuscript is clearly written and the experimentation is convincing. I have just a few points:

2.1. The association of the PRPH splice variant to SNAP is convincing. The link between SNAP and neuropathy might need more clarification. After all, as this reviewer understands it, the diagnostic criteria of peripheral neuropathy do not include SNAP. This should be clarified for an audience which does not consist of neurophysiologists. In other words what is the role of SNAP in peripheral neuropathy?

Response (2.1.): The reviewer is correct in that clinical diagnostic criteria for common forms of peripheral neuropathy (PN), do not necessarily include measuring SNAP. Indeed, NC results are not part of the standardized UENS (Utah Early Neuropathy Scale) clinical criteria used in this article to determine PN status of the re-called carriers and controls. That said, the lack of a gold standard for diagnosing PN has long been a dilemma for clinicians, and most comprehensive guidelines for diagnosing PN do recommend objective measures of nerve function, such as nerve conduction, quantitative sensory testing (QST), or both (*Horowitz, S.H. Criteria for the diagnosis of peripheral neuropathies. Occup Environ Med 59, 425-6 (2002), PMID: 12107287*). We measured both; the results per rs73112142 genotype shown in Supplementary Table 5.

Guided by the reviewer’s comment and question on the role of SNAP in PN, we have made the following revisions to the manuscript for increased reader clarity, as follows:

- a) We revised the sentence on page 3 starting in line 67: “Nerve Conduction (NC) studies probe electrophysiological properties of nerves and are used **to aid in diagnosing and classifying PN**” (from previously ...used to diagnose and classify PN),
- b) Added the sentence: “Given possible axonal pathology of the sural nerve as suggested by the association of rs73112142-A with SNAP, but not SNCV“, to the results section starting in line 212 (page 9),
- c) Added the sentence "As subjects were without known PN risk factors, the scale most sensitive to early neuropathy detection (UENS) was used to classify PN status.", to the same section starting in line 222 (page 9), and
- d) Added the statement: “While NC results are not part of the UENS criteria.. “, to the sentence starting in line 230(page 9).

2.2. Related to point (1), on lines 116-117 the sentence is written in a way that the reader understands that a PN diagnosis is met if SNAP and SNCV are below reference values. Is that really so? Isn’t PN after all a clinical diagnosis?

Response (2.2.): As per our response to comment 2.1 above, we agree with the reviewer, that diagnostic criteria for PN require evaluation of signs and symptoms, and NC studies alone do not suffice. That said, NC measures using the same device as we used in our study, have been tested against clinical examination of PN (diabetic polyneuropathy) and exhibited 90.5% sensitivity, 86.1% specificity, 79.2% positive predictive value and 93.9% negative predictive value (*Chatzikosma, G. et al. Evaluation of sural nerve automated nerve conduction study in the diagnosis of peripheral neuropathy in patients with type 2 diabetes mellitus. Arch Med Sci 12, 390-3 (2016), PMID: 27186185*). Therefore, we initially used these NC screening criteria as a proxy for PN in the larger, unexamined sample (as "screen-positive PN"). However, as we did not find significant associations with this phenotype, and it’s mention leads to a potential misunderstanding for readers as identified by the reviewer, we conclude that it is clearer for readers of the paper, to omit altogether this proxy

PN phenotype from our manuscript. It is therefore now removed from Table 1, results section, and the Manhattan plot from Supplementary figure 2.

2.3. The study using PRPH minor allele homozygotes is a brilliant way to understand the consequences of the variant and having an infrastructure where such cases can be recalled is unique. However, understandably, the sample is small. Setting the diagnosis of PN remains slightly unclear to the reader. On line 218 it is stated that symptoms are mild. This raises the question, do the findings reach the diagnosis of PN or should they be called "mild PN like symptoms". Please clarify. This is especially unclear as the authors stress on the younger age of onset of PN symptoms. If the symptoms are very mild, what is the specifics of evaluating the age of onset.

Response (2.3.): We thank the reviewer for recognizing the strengths of our approach in recruiting individuals based on genotype for in-depth phenotyping. We regret that the reviewer finds the description of how the PN classification was set, unclear and hope that by the revisions detailed in response 2.1 will have clarified this point.

We agree with the reviewer that the PN symptoms reported by carriers classified with PN were mild. However, while symptoms were not serious enough for affected carriers (who were otherwise in good health, without known PN risk factors), to seek medical attention, the standardized PN assessment scales showed that especially homozygote carriers had higher scores, reflecting more PN signs and symptoms, than non-carriers (with no significant age difference between carrier groups) (Supplementary Table 5). Subjective text data obtained by clinical examiners, (not included in manuscript), also suggested that affected carriers found their symptoms (especially numbness) annoying, however some also stating that they had always been this way and had were not concerned. Further studies of the PN associated with this variant will be important to determine the clinical prognosis, especially if carriers are faced with other PN risks, which was not the case in this study.

With respect to the reviewer's question about age of onset, for each positive PN symptom reported, participants were asked at what age they could first recall experiencing given symptom, and age of onset was assessed according to their recall of their earliest relevant PN symptom. This is now more clearly expressed in the results chapter (line 229) (age of onset changed to "...they recalled being significantly younger when first experiencing their PN symptoms". The response: "I have always been this way" was *e.g.* assigned age of onset at 10 yrs.

2.4. On line 222 there is a reference to Supplemental Table 6. At least this reviewer cannot find Suppl Table 6.

Response (2.4.): We thank the reviewer for pointing this out, the reference should be to Supplemental Table 5 and we have corrected this error.

2.5. It would be helpful if the authors would report what they observe in UKBB. Is the variant imputed in the UKBB data. Moreover, it would be good to discuss the possibilities for replicating the presented results. Was there an attempt for this?

Response (2.5.): The variant is imputed in the UKBbiobank data, but is about three times rarer in the UK than in Iceland, or MAF = 0.47%. Thus, the homozygous genotypic frequency is close to ten times rarer in the UK compared to Iceland. As stated in our response to Reviewer #1 in 1.2 above, we looked for comparably phenotyped samples in other populations for replication of our findings, but without success. Even the comprehensive phenotyping performed by the UK Biobank does not

include nerve conduction measures, hence, we were not able to replicate our findings using these data.

2.6. On line 223 the authors state: “..we observe an increased risk of early-onset, predominantly sensory peripheral polyneuropathy”. The p-value is nominally significant, so maybe the “topic sentence” could be somewhat tuned down. Is the p-value corrected for multiple testing?

Response (2.6.): The association with the variant was identified in a GWAS of SNAP, where we fully accounted for multiple testing. Here, we are simply testing the variant of interest for association with risk of PN (see response to 2.1). In the manuscript we state the following:

“Under an additive model, the association (OR = 4.5, (95%CI = 1.3 - 14.7), $P = 0.015$, $N = 69$) is driven primarily by homozygotes. Under a recessive model, the risk for homozygotes is 26-fold (OR = 26.1 (95%CI = 3.1 - 218.1), $P = 2.6 \times 10^{-3}$, $N = 9$)”.

Here the observed P -value is under 0.025 (0.05/2) for both models tested and comfortably so for the recessive model. Furthermore, we state in the discussion that further studies of the clinical impact and prognosis of the reported observations are called for.

Reviewer #3 (Remarks to the Author):

3. In this paper, the authors have performed a GWAS of sural nerve conduction amplitude (SNAP) and velocity (SNCV) in 7,045 Icelanders. They found that a splice donor variant in the gene encoding the intermediate filament protein, peripherin associates with decreased SNAP, but not SNCV. Defective splicing due to this variant yields two truncated frameshift peripherin proteins, with a truncate rod domain as a result. Overexpression of the splice variant's isoform in a cell line devoid of intermediate filaments showed that the variants do not allow formation of the normal filamentous structure of peripherin. Overall, this is an interesting study that implicates peripherin dysfunction in sural nerve conduction amplitude, which has previously not been observed. I have a few comments and/or questions.

3.1. The most important one has to do with the mechanism by which this variance would cause problems. The results of the transfection studies suggest that the splice variant isoforms cannot form filaments on their own, but the more relevant question is whether this results in a dominant negative failure of any intermediate filaments to form. This could easily be studied by co-transfecting wild-type peripherin with the variants.

Response (3.1.): This intriguing question raised by the reviewer addresses the detailed molecular mechanisms leading to the formation of peripherin inclusions. It has been reported previously, that an aggregation-prone C-terminally truncated peripherin splice variant, named Per-28, is sequestered within intracellular inclusions (McLean et al. *Experimental Neurology* 2014; PMID 24907400), and selectively interacts with other filamentous proteins as its expression leads to an almost total collapse in Vimentin (another class-III IF) networks. An analogous observation was made by Leung et al. (Leung et al. *Brain Pathology* 2006; PMID 15446584), whereby an aggregate-prone mutant peripherin co-localized with neurofilament light (NFL) protein in compact aggregates. Interestingly, Leung et al. additionally reported that co-expression of WT peripherin lead to a partial rescue of peripherin aggregates.

Our observations are consistent with an additive effect of the variant protein as we observed a clear dose-dependent genotype effect on SNAP, with homozygote carriers having around 3-fold lower SNAP than heterozygotes, when both are compared to non-carriers. If the variant was acting in a dominant negative manner then the effect should be comparable between homo- and heterozygote

carriers. Our data therefore argue against a dominant negative effect that would lead to a complete failure of intermediate filamentous network formation.

Since we still do not have a detailed understanding of how variant forms of peripherin cause aggregation/inclusion formation, it presents both an important and exciting line of investigation. Among the outstanding questions, in addition to and including the points raised by Reviewer #3, is how the variant peripherin affects the formation of the network of filaments containing the neurofilament triplets; NFL, NFM and NFH, in neural cells. As peripherin has been demonstrated to be a subunit of these complexes, it is important that such analyses should be performed within the physiological stoichiometry between the separate subunits of the complexes, preferably in the correct cellular context. However, these and further detailed investigations into the kinetics of filament network formation in the presence of variant peripherin protein lies beyond the scope of the current study.

We thank Reviewer #3 for raising these points, and have incorporated them into the discussion section (pages 10-11) of the manuscript.

3.2. A second possibility is suggested by the study of peripherin expression in adipose tissues, which showed that transcripts from the mutated allele were 41% fewer than those from the wild-type allele. This suggest that the results could be because of lower levels of peripherin. Can the authors comment on this.

Response (3.2.): Since submitting this manuscript, our Icelandic RNAseq dataset has been expanded, both in terms of an increased number of analyzed adipose samples (now N=749), increased sequencing depth of heterozygous carriers of the variant allele (~5-fold), and one heterozygous carrier sample was added. We therefore reanalyzed the RNAseq data (hence, adding an author involved in conducting these analyses, Pall Melsted), finding basically the same results, although some numbers have changed. All RNA-seq results and associated figures (Now Figure 4, page 28-30, presented in 3 panels; 4A, 4B and 4C), have been updated in the revised manuscript and the figure legends and text modified for clarity.

Regarding the peripherin transcripts in the heterozygous carriers of the variant allele, we observe the following:

- a) From the variant allele we observe that all transcripts generated are predicted to produce truncated peripherin protein.
- b) The *PRPH* transcripts are reduced in adipose tissue from heterozygotes compared to wild type (WT) (Effect = -22.7% (95%CI: -33.0 to -10.9%), $P = 0.0004$) and this reduction is caused by fewer variant allele transcripts, compared to those from the WT allele (35% vs 75%). This suggests that the abnormal transcripts undergo partial nonsense mediated decay that translates into reduction in total *PRPH* transcripts. Extrapolating from this, homozygote carriers should therefore not express any correctly spliced transcripts.

Since the reduction in the generation of full-length protein is, both in heterozygotes and homozygotes, accompanied by the presence of the truncated proteins we cannot state if SNAP reduction in the sural nerve is because of the loss of full length protein or the presence of the truncated proteins, or a combination of both. We have incorporated these updated analyses in the results section (page 6, lines 147- 169), the discussion section of the manuscript (page 10-11, lines 253-267), and a detailed description of the RNA seq analysis in the methods section (page 16, starting in line 401).

3.3. The results mentioned in item 2 (3.2), could be better described and explained. Since it is only shown in the supplementary figures (why?), the figure legend lacks sufficient

detail and the experimental details are also not described. The fact that peripherin is expressed in adipose tissues is also a bit of a surprise to me. What levels are they expressed at, and in what cells? Peripherin is thought to be primarily in neurons.

Response (3.3.): Based on both GTex (RNA) and the Human protein atlas (RNA and protein), peripherin appears to be expressed in various tissues other than neuronal tissue, including adipose tissue. Regarding the level of expression in our RNA adipose sample that is from subcutaneous fat, 40% of the expressed genes have higher RNA expression than *PRPH*.

The greatest value of these adipose expression data to this study is the ability to determine the effect of the variant allele on peripherin transcripts, both their structures and levels.

As suggested by the reviewer, in the revised manuscript we have moved the supplementary figure describing RNA expression in adipose tissue to the main section of the manuscript (now Main Figure 4 in three panels (A,B,C, see previous response), with detailed figure legends).

3.4. Minor quibble: the SW13 Vim- cells are said to be devoid of filaments. To be precise, they are devoid of intermediate filaments, there are multiple other filament (microfilaments/microtubules) present in these cells.

Response (3.4.): We thank Reviewer #3 for pointing this out and have revised the abstract (line 33) accordingly.

Furthermore, in response to the reviewer's overall comments, we revised the abstract to emphasize that through RNA *and* protein studies, we demonstrated that the splice-donor variant rs73112142 (c.996+1G>A) is indeed a LoF (loss-of-function) variant.

Additional revisions:

- Authors: Pall Melsted added to author list, in light of role in additional RNA-seq analysis.
- Abstract: We made slight changes so that it would be clear from the abstract that we performed both RNA and protein studies to demonstrate that the variant associating with Sural NC amplitude (SNAP) leads to loss-of-function (LoF) of *PRPH*. To maintain the word count <150 words, we omitted the previous first sentence of the abstract ("*Peripheral neuropathy is a term applied to a group of peripheral nerve diseases for which limited treatment exists*").
- Line 66: The word "sizes" (in sample sizes), was unnecessary and omitted.
- Lines 85-86: Errors corrected to reflect accurate N counts presented in Table 1.
- Line 90: BMI is now spelled out (body mass index).
- Lines 131-132: To clarify sentence in text, change from: "We also observed a clear dose-dependent genotype effect.....", to: "In addition to the clear dose-dependent genotype effect on SNAP, we also observe.."
- Line 138: To more accurately represent the findings discussed in the section, the title: "The splice-donor variant generates two truncated PRPH proteins" was changed to: "Defective splicing generates two truncated frameshift protein isoforms"
- Line 306: Changed sentence from "...previously linked to neurodegenerative disease". In light of the weak evidence of association with neurodegeneration (OMIM #170710), we changed this to ".....without previous phenotype associations".
- Added a key reference (nr 10) regarding interpretation of NC results in classifying PN (*Tankisi, H. et al. Pathophysiology inferred from electrodiagnostic nerve tests and classification of polyneuropathies. Suggested guidelines. Clin Neurophysiol 116, 1571-80 (2005)*).
- Some minor wording changes were also made to improve readability of the manuscript. Although not listed here, they are highlighted in the revised manuscript.

In conclusion, we hope that the Editors and Reviewers find our responses and adjustments satisfactory and that the revised manuscript fulfils Nature Communication's requirements for publication. We hope that this first unequivocal genome-wide association of a high-impact variant in *PRPH* with sural nerve conduction amplitude and risk of a mild form of peripheral neuropathy congruent with axonal loss, will fuel subsequent studies in other populations and extend the scientific community's understanding of the biological underpinnings of peripheral nerve function and the role of genetics in complex neuropathies.

Thank you for your continued interest in our manuscript.

Sincerely,

Corresponding authors Gyda Bjornsdottir and Kari Stefansson,
deCODE Genetics /Amgen
Reykjavik, Iceland

Reviewer #1 (Remarks to the Author):

I thank the Authors for their careful and thoughtful response to my (and the other Reviewer) comments.

Their responses and adjustments happily satisfy my questions and requests and have improved an already excellent report.

Reviewer #2 (Remarks to the Author):

The authors have adequately addressed all critiques raised by this reviewer. No further comments.

Reviewer #3 (Remarks to the Author):

The authors have made adequate revisions to the manuscript.